# Predicting Useful Neighborhoods
# for Lazy Local Learning

**Aron Yu**
University of Texas at Austin
aron.yu@utexas.edu

**Kristen Grauman**
University of Texas at Austin
grauman@cs.utexas.edu

## Abstract

Lazy local learning methods train a classifier "on the fly" at test time, using only a subset of the training instances that are most relevant to the novel test example. The goal is to tailor the classifier to the properties of the data surrounding the test example. Existing methods assume that the instances most useful for building the local model are strictly those closest to the test example. However, this fails to account for the fact that the success of the resulting classifier depends on the full *distribution* of selected training instances. Rather than simply gathering the test example's nearest neighbors, we propose to predict the subset of training data that is jointly relevant to training its local model. We develop an approach to discover patterns between queries and their "good" neighborhoods using large-scale multi-label classification with compressed sensing. Given a novel test point, we estimate both the composition and size of the training subset likely to yield an accurate local model. We demonstrate the approach on image classification tasks on SUN and aPascal and show its advantages over traditional global and local approaches.

## 1 Introduction

Many domains today—vision, speech, biology, and others—are flush with data. Data availability, combined with recent large-scale annotation efforts and crowdsourcing developments, have yielded labeled datasets of unprecedented size. Though a boon for learning approaches, large labeled datasets also present new challenges. Beyond the obvious scalability concerns, the diversity of the data can make it difficult to learn a single global model that will generalize well. For example, a standard binary *dog* classifier forced to simultaneously account for the visual variations among hundreds of dog breeds may be "diluted" to the point it falls short in detecting new dog instances. Furthermore, with training points distributed unevenly across the feature space, the model capacity required in any given region of the space will vary. As a result, if we train a single high capacity learning algorithm, it may succeed near parts of the decision boundary that are densely populated with training examples, yet fail in poorly sampled areas of the feature space.

Local learning methods offer a promising direction to address these challenges. Local learning is an instance of "lazy learning", where one defers processing of the training data until test time. Rather than estimate a single global model from all training data, local learning methods instead focus on a subset of the data most relevant to the particular test instance. This helps learn fine-grained models tailored to the new input, and makes it possible to adjust the capacity of the learning algorithm to the local properties of the data [5]. Local methods include classic nearest neighbor classification as well as various novel formulations that use only nearby points to either train a model [2, 3, 5, 13, 29] or learn a feature transformation [8, 9, 15, 25] that caters to the novel input.

A key technical question in local learning is how to determine which training instances are relevant to a test instance. All existing methods rely on an important core assumption: that the instances most *useful* for building a local model are those that are *nearest* to the test example. This assumption is well-motivated by the factors discussed above, in terms of data density and intra-class variation.

Furthermore, identifying training examples solely based on proximity has the appeal of permitting specialized similarity functions (whether learned or engineered for the problem domain), which can be valuable for good results, especially in structured input spaces.

On the other hand, there is a problem with this core assumption. By treating the individual nearness of training points as a metric of their utility for local training, existing methods fail to model how those training points will actually be employed. Namely, the relative success of a locally trained model is a function of the entire *set* or *distribution* of the selected data points—not simply the individual pointwise nearness of each one against the query. In other words, the ideal target subset consists of a set of instances that together yield a good predictive model for the test instance.

Based on this observation, we propose to learn the properties of a "good neighborhood" for local training. Given a test instance, the goal is to predict which subset of the training data should be enlisted to train a local model on the fly. The desired prediction task is non-trivial: with a large labeled dataset, the power set of candidates is enormous, and we can observe relatively few training instances for which the most effective neighborhood is known. We show that the problem can be cast in terms of large-scale multi-label classification, where we learn a mapping from an individual instance to an indicator vector over the entire training set that specifies which instances are jointly useful to the query. Our approach maintains an inherent bias towards neighborhoods that are local, yet makes it possible to discover subsets that (i) deviate from a strict nearest-neighbor ranking and (ii) vary in size.

The proposed technique is a general framework to enhance local learning. We demonstrate its impact on image classification tasks for computer vision, and show its substantial advantages over existing local learning strategies. Our results illustrate the value in estimating the size and composition of discriminative neighborhoods, rather than relying on proximity alone.

## 2 Related Work

**Local learning algorithms** Lazy local learning methods are most relevant to our work. Existing methods primarily vary in how they exploit the labeled instances nearest to a test point. One strategy is to identify a fixed number of neighbors most similar to the test point, then train a model with only those examples (e.g., a neural network [5], SVM [29], ranking function [3, 13], or linear regression [2]). Alternatively, the nearest training points can be used to learn a transformation of the feature space (e.g., Linear Discriminant Analysis); after projecting the data into the new space, the model is better tailored to the query's neighborhood properties [8, 9, 15, 25]. In *local selection* methods, strictly the subset of nearby data is used, whereas in *locally weighted* methods, all training points are used but weighted according to their distance [2]. All prior methods select the local neighborhood based on proximity, and they typically fix its size. In contrast, our idea is to *predict* the set of training instances that will produce an effective discriminative model for a given test instance.

**Metric learning** The question "what is relevant to a test point?" also brings to mind the metric learning problem. Metric learning methods optimize the parameters of a distance function so as to best satisfy known (dis)similarity constraints between training data [4]. Most relevant to our work are those that learn *local* metrics; rather than learn a single global parameterization, the metric varies in different regions of the feature space. For example, to improve nearest neighbor classification, in [11] a set of feature weights is learned for each individual training example, while in [26, 28] separate metrics are trained for clusters discovered in the training data.

Such methods are valuable when the data is multi-modal and thus ill-suited by a single global metric. Furthermore, one could plug a learned metric into the basic local learning framework. However, we stress that learning what a good *neighbor* looks like (metric learning's goal) is distinct from learning what a good *neighborhood* looks like (our goal). Whereas a metric can be trained with pairwise constraints indicating what should be near or far, jointly predicting the instances that ought to compose a neighborhood requires a distinct form of learning, which we tackle in this work.

**Hierarchical classification** For large multi-class problems, hierarchical classification approaches offer a different way to exploit "locality" among the training data. The idea is to assemble a tree of decision points, where at each node only a subset of labels are considered (e.g., [6, 12, 21]). Such methods are valuable for reducing computational complexity at test time, and broadly speaking they share the motivation of focusing on finer-grained learning tasks to improve accuracy. However,

otherwise the work is quite distant from our problem. Hierarchical methods precompute groups of labels to isolate in classification tasks, and apply the same classifiers to all test instances; lazy local learning predicts at test time what set of training instances are relevant for each novel test instance.

**Weighting training instances**  Our problem can be seen as deciding which training instances to "trust" most. Various scenarios call for associating weights with training instances such that some influence the learned parameters more than others. For example, weighted instances can reflect label confidences [27], help cope with imbalanced training sets [24], or resist the influence of outliers [20]. However, unlike our setting, the weights are given at training time and they are used to create a single global model. Methods to *estimate* the weights per example arise in domain adaptation, where one aims to give more weight to source domain samples distributed most like those in the target domain [14, 17, 18]. These are non-local, offline approaches, whereas we predict useful neighborhoods in an online, query-dependent manner. Rather than close the mismatch between a source and target domain, we aim to find a subset of training data amenable to a local model.

**Active learning**  Active learning [23] aims to identify informative *unlabeled* training instances, with the goal of minimizing labeling effort when training a single (global) classifier. In contrast, our goal is to ignore those labeled training points that are irrelevant to a particular novel instance.

## 3 Approach

We propose to predict the set of training instances which, for a given test example, are likely to compose an effective neighborhood for local classifier learning. We use the word "neighborhood" to refer to such a subset of training data—though we stress that the optimal subset need not consist of strictly rank-ordered nearest neighbor points.

Our approach has three main phases: (i) an offline stage where we generate positive training neighborhoods (Sec. 3.1), (ii) an offline stage where we learn a mapping from individual examples to their useful neighborhoods (Sec. 3.2), and (iii) an online phase where we apply the learned model to infer a novel example's neighborhood, train a local classifier, and predict the test label (Sec. 3.3).

### 3.1 Generating training neighborhoods

Let $\mathcal{T} = \{(\boldsymbol{x}_1, c_1), \ldots, (\boldsymbol{x}_M, c_M)\}$ denote the set of $M$ category-labeled training examples. Each $\boldsymbol{x}_i \in \Re^d$ is a vector in some $d$-dimensional feature space, and each $c_i \in \{1, \ldots, C\}$ is its target category label. Given these examples, we first aim to generate a set of *training neighborhoods*, $\mathcal{N} = \{(\boldsymbol{x}_{n_1}, \boldsymbol{y}_{n_1}), \ldots, (\boldsymbol{x}_{n_N}, \boldsymbol{y}_{n_N})\}$. Each training neighborhood $(\boldsymbol{x}_{n_i}, \boldsymbol{y}_{n_i})$ consists of an individual instance $\boldsymbol{x}_{n_i}$ paired with a set of training instance indices capturing its target "neighbors", the latter being represented as a $M$-dimensional indicator vector $\boldsymbol{y}_{n_i}$. If $\boldsymbol{y}_{n_i}(j) = 1$, this means $\boldsymbol{x}_j$ appears in the target neighborhood for $\boldsymbol{x}_{n_i}$. Otherwise, $\boldsymbol{y}_{n_i}(j) = 0$. Note that the dimensionality of this target indicator vector is $M$, the number of total available training examples. We will generate $N$ such pairs, where typically $N \ll M$.

As discussed above, there are very good motivations for incorporating nearby points for local learning. Indeed, we do not intend to eschew the "locality" aspect of local learning. Rather, we start from the premise that points near to a query are likely relevant—but relevance is not necessarily preserved purely by their rank order, nor must the best local set be within a fixed radius of the query (or have a fixed set size). Instead, we aim to generalize the locality concept to *jointly* estimate the members of a neighborhood such that *taken together* they are equipped to train an accurate query-specific model.

With these goals in mind, we devise an empirical approach to generate the pairs $(\boldsymbol{x}_{n_i}, \boldsymbol{y}_{n_i}) \in \mathcal{N}$. The main idea is to sample a series of candidate neighborhoods for each instance $\boldsymbol{x}_{n_i}$, evaluate their relative success at predicting the training instance's label, and record the best candidate.

Specifically, for instance $\boldsymbol{x}_{n_i}$, we first compute its proximity to the $M - 1$ other training images in the feature space. (We simply apply Euclidean distance, but a task-specific kernel or learned metric could also be used here.) Then, for each of a series of possible neighborhood sizes $\{k_1, \ldots, k_K\}$, we sample a neighborhood of size $k$ from among all training images, subject to two requirements: (i) points nearer to $\boldsymbol{x}_{n_i}$ are more likely to be chosen, and (ii) the category label composition within the neighborhood set is balanced. In particular, for each possible category label $1, \ldots, C$ we sample $\frac{k}{C}$ training instances without replacement, where the weight associated with an instance is inversely

related to its (normalized) distance to $\boldsymbol{x}_{n_i}$. We repeat the sampling $S$ times for each value of $k$, yielding $K \times S$ candidates per instance $\boldsymbol{x}_{n_i}$.

Next, for each of these candidates, we learn a local model. Throughout we employ linear support vector machine (SVM) classifiers, both due to their training efficiency and because lower capacity models are suited to the sparse, local datasets under consideration; however, kernelized/non-linear models are also possible.[1] Note that any number of the $K \times S$ sampled neighborhoods may yield a classifier that correctly predicts $\boldsymbol{x}_{n_i}$'s category label $c_{n_i}$. Thus, to determine which among the successful classifiers is best, we rank them by their prediction confidences. Let $p_s^k(\boldsymbol{x}_{n_i}) = P(c_{n_i}|\boldsymbol{x}_{n_i})$ be the posterior estimated by the $s$-th candidate classifier for neighborhood size $k$, as computed via Platt scaling using the neighborhood points. To automatically select the best $k$ for instance $\boldsymbol{x}_{n_i}$, we average these posteriors across all samples per $k$ value, then take the one with the highest probability: $k^* = \arg\max_k \frac{1}{S} \sum_{s=1}^{S} p_s^k(\boldsymbol{x}_{n_i})$. The averaging step aims to smooth the estimated probability using the samples for that value of $k$, each of which favors near points but varies in its composition. Finally, we obtain a single neighborhood pair $(\boldsymbol{x}_{n_i}, \boldsymbol{y}_{n_i})$, where $\boldsymbol{y}_{n_i}$ is the indicator vector for the neighborhood sampled with size $k^*$ having the highest posterior $p_s^{k^*}$.

In general we can expect higher values of $S$ and denser samplings of $k$ to provide best results, though at a correspondingly higher computational cost during this offline training procedure.

### 3.2 Learning to predict neighborhoods with compressed sensing

With the training instance-neighborhood pairs in hand, we next aim to learn a function capturing their relationships. This function must estimate the proper neighborhood for novel test instances. We are faced with a non-trivial learning task. The most straightforward approach might be to learn a binary decision function for each $\boldsymbol{x}_i \in \mathcal{T}$, trained with all $\boldsymbol{x}_{n_j}$ for which $\boldsymbol{y}_{n_j}(i) = 1$ as positives. However, this approach has several problems. First, it would require training $M$ binary classifiers, and in the applications of interest $M$—the number of all available category-labeled examples—may be very large, easily reaching the millions. Second, it would fail to represent the dependencies between the instances appearing in a single training neighborhood, which ought to be informative for our task. Finally, it is unclear how to properly gather *negative* instances for such a naive solution.

Instead, we pose the learning task as a large-scale multi-label classification problem. In multi-label classification, a single data point may have multiple labels. Typical examples include image and web page tagging [16, 19] or recommending advertiser bid phrases [1]. In our case, rather than predict which labels to associate with a novel example, we want to predict which training instances belong in its neighborhood. This is exactly what is encoded by the target indicator vectors defined above, $\boldsymbol{y}_{n_i}$. Furthermore, we want to exploit the fact that, compared to the number of all labeled training images, the most useful local neighborhoods will contain relatively few examples.

Therefore, we adopt a large-scale multi-label classification approach based on compressed sensing [19] into our framework. With it, we can leverage sparsity in the high-dimensional target neighborhood space to efficiently learn a prediction function that jointly estimates all useful neighbors. First, for each of the $N$ training neighborhoods, we project its $M$-dimensional neighborhood vector $\boldsymbol{y}_{n_i}$ to a lower-dimensional space using a random transformation: $\boldsymbol{z}_{n_i} = \boldsymbol{\phi}\, \boldsymbol{y}_{n_i}$, where $\boldsymbol{\phi}$ is a $D \times M$ random matrix, and $D$ denotes the compressed indicators' dimensionality. Then, we learn regression functions to map the original features to these projected values $\boldsymbol{z}_{n_1}, \ldots, \boldsymbol{z}_{n_N}$ as targets. That is, we obtain a series of $D \ll M$ regression functions $f_1, \ldots, f_D$ minimizing the loss in the compressed indicator vector space. Given a novel instance $\boldsymbol{x}_q$, those same regression functions are applied to map to the reduced space, $[f_1(\boldsymbol{x}_q), \ldots, f_D(\boldsymbol{x}_q)]$. Finally, we predict the complete indicator vector by recovering the $M$-dimensional vector using a standard reconstruction algorithm from the compressed sensing literature.

We employ the Bayesian multi-label compressed sensing framework of [19], since it unifies the regression and sparse recovery stages, yielding accurate results for a compact set of latent variables. Due to compressed sensing guarantees, an $M$-dimensional indicator vector with $l$ nonzero entries can be recovered efficiently using $D = O(l \log \frac{M}{l})$ [16].

### 3.3 Inferring the neighborhood for a novel example

All processing so far is performed offline. At test time, we are given a novel example $x_q$, and must predict its category label. We first predict its neighborhood using the compressed sensing approach overviewed in the previous section, obtaining the $M$-dimensional vector $\hat{y}_q$. The entries of this vector are real-valued, and correspond to our relative confidence that each category-labeled instance $x_i \in \mathcal{T}$ belongs in $x_q$'s neighborhood.

Past multi-label classification work focuses its evaluation on the precision of (a fixed number of) the top few most confident predictions and the raw reconstruction error [16, 19], and does not handle the important issue of how to truncate the values to produce hard binary decisions. In contrast, our setting demands that we extract both the neighborhood size estimate as well as the neighborhood composition from the estimated real-valued indicator vector.

To this end, we perform steps paralleling the training procedure defined in Sec. 3.1, as follows. First, we use the sorted confidence values in $\hat{y}_q$ to generate a series of candidate neighborhoods of sizes varying from $k_1$ to $k_K$, each time ensuring balance among the category labels. That is, for each $k$, we take the $\frac{k}{C}$ most confident training instances per label. Recall that all $M$ training instances referenced by $\hat{y}_q$ have a known category label among $1, \ldots, C$. Analogous to before, we then apply each of the $K$ candidate *predicted* neighborhoods in turn to train a local classifier. Of those, we return the category label prediction from the classifier with the most confident decision value.

Note that this process automatically selects the neighborhood size $k$ to apply for the novel input. In contrast, existing local learning approaches typically manually define this parameter and fix it for all test examples [5, 8, 13, 15, 29]. Our results show that approach is sub-optimal; not only does the most useful neighborhood deviate from the strict ranked list of neighbors, it also varies in size.

We previously explored an alternative approach for inference, where we directly used the confidences in $\hat{y}_q$ as weights in an importance-weighted SVM. That is, for each query, we trained a model with all $M$ data points, but modulated their influence according to the soft indicator vector $\hat{y}_q$, such that less confident points incurred lower slack penalties. However, we found that approach inferior, likely due to the difficulty in validating the slack scale factor for all training instances (problematic in the local learning setting) as well as the highly imbalanced datasets we tackle in the experiments.

### 3.4 Discussion

While local learning methods strive to improve accuracy over standard global models, their lazy use of training data makes them more expensive to apply. This is true of any local approach that needs to compute distances to neighbors and train a fresh classifier online for each new test example. In our case, using Matlab, the run-time for processing a single novel test point can vary from 30 seconds to 30 minutes. It is dominated by the compressed sensing reconstruction step, which takes about 80% of the computation time and is highly dependent on the complexity of the trained model. One could improve performance by using approximate nearest neighbor methods to sort $\mathcal{T}$, or pre-computing a set of representative local models. We leave these implementation improvements as future work.

The offline stages of our algorithm (Secs. 3.1 and 3.2) require about 5 hours for datasets with $M = 14,000$, $N = 2,000$, $d = 6,300$, and $D = 2,000$. The run-time is dominated by the SVM evaluation of $K \times S$ candidate training neighborhoods on the $N$ images, which could be performed in parallel. The compressed sensing formulation is quite valuable for efficiency here; if we were to instead naively train $M$ independent classifiers, the offline run-time would be on the order of days.

We found that building category-label balance into the training and inference algorithms was crucial for good results when dealing with highly imbalanced datasets. Earlier versions of our method that ignored label balance would often predict neighborhoods with only the same label as the query. Local methods typically handle this by reverting to a nearest neighbor decision. However, as we will see below, this can be inferior to explicitly learning to identify a local *and* balanced neighborhood, which can be used to build a more sophisticated classifier (like an SVM).

Finally, while our training procedure designates a single neighborhood as the prediction target, it is determined by a necessarily limited sample of candidates (Sec. 3.1). Our confidence ranking step accounts for the differences between those candidates that ultimately make the same label prediction. Nonetheless, the non-exhaustive training samples mean that slight variations on the target vectors

may be equally good in practice. This suggests future extensions to explicitly represent "missing" entries in the indicator vector during training or employ some form of active learning.

## 4 Experiments

We validate our approach on an array of binary image classification tasks on public datasets.

**Datasets** We consider two challenging datasets with visual attribute classification tasks. The SUN Attributes dataset [22] (**SUN**) contains 14,340 scene images labeled with binary attributes of various types (e.g., materials, functions, lighting). We use all images and randomly select 8 attribute categories. We use the 6,300-dimensional HOG $2 \times 2$ features provided by the authors, since they perform best for this dataset [22]. The **aPascal** training dataset [10] contains 6,440 object images labeled with attributes describing the objects' shapes, materials, and parts. We use all images and randomly select 6 attribute categories. We use the base features from [10], which include color, texture, edges, and HOG. We reduce their dimensionality to 200 using PCA. For both datasets, we treat each attribute as a separate binary classification task ($C = 2$).

**Implementation Details** For each attribute, we compose a test set of 100 randomly chosen images (balanced between positives and negatives), and use all other images for $\mathcal{T}$. This makes $M = 14,240$ for SUN and $M = 6,340$ for aPascal. We use $N = 2,000$ training neighborhoods for both, and set $D = \{2000, 1000\}$ for SUN and aPascal, roughly 15% of their original label indicator lengths. Generally higher values of $D$ yield better accuracy (less compression), but for a greater expense. We fix the number of samples $S = 100$, and consider neighborhood sizes from $k_1 = 50$ and $k_K = 500$, in increments of 10 to 50.

**Baselines and Setup** We compare to the following methods: (1) **Global**: for each test image, we apply the same global classifier trained with all $M$ training images; (2) **Local**: for each test image, we apply a classifier trained with only its nearest neighbors, as measured with Euclidean distance on the image features. This baseline considers a series of $k$ values, like our method, and independently selects the best $k$ per test point according to the confidence of the resulting local classifiers (see Sec. 3.3). (3) **Local+ML**: same as Local, except the Euclidean distance is replaced with a learned metric. We apply the ITML metric learning algorithm [7] using the authors' public code.

Global represents the default classification approach, and lets us gauge to what extent the classification task requires local models at all (e.g., how multi-modal the dataset is). The two Local baselines represent the standard local learning approach [3, 5, 13, 15, 25, 29], in which proximal data points are used to train a model per test case, as discussed in Sec. 2. By using proximity instead of $\hat{\boldsymbol{y}}_q$ to define neighborhoods, they isolate the impact of our compressed sensing approach.

All results reported for our method and the Local baselines use the automatically selected $k$ value per test image (cf. Sec. 3.3), unless otherwise noted. Each local method independently selects its best $k$ value. All methods use the exact same image features and train linear SVMs, with the cost parameter cross-validated based on the Global baseline. To ensure the baselines do not suffer from the imbalanced data, we show results for the baselines using both *balanced* (B) and *unbalanced* (U) training sets. For the balanced case, for Global we randomly downsample the negatives and average results over 10 such runs, and for Local we gather the nearest $\frac{k}{2}$ neighbors from each class.

**SUN Results** The SUN attributes are quite challenging classification tasks. Images within the same attribute exhibit wide visual variety. For example, the attribute "eating" (see Fig. 1, top right) is positive for any images where annotators could envision eating occurring, spanning from an restaurant scene, to home a kitchen, to a person eating, to a banquet table close-up. Furthermore, the attribute may occupy only a portion of the image (e.g., "metal" might occupy any subset of the pixels). It is exactly this variety that we expect local learning may handle well.

Table 1 shows the results on SUN. Our method outperforms all baselines for all attributes. Global benefits from a balanced training set (B), but still underperforms our method (by 6 points on average). We attribute this to the high intra-class variability of the dataset. Most notably, conventional Local learning performs very poorly—whether or not we enforce balance. (Recall that the test sets are always balanced, so chance is 0.50.) Adding metric learning to local (Local+ML) improves things only marginally, likely because the attributes are not consistently localized in the image. We also implemented a *local* metric learning baseline that clusters the training points then learns a met-

| Attribute | Global | | Local | | Local+ML | | Ours | Local | Local+ML | Ours | Ours |
|---|---|---|---|---|---|---|---|---|---|---|---|
| | B | U | B | U | B | U | | | $k = 400$ | | **Fix-$k$\*** |
| hiking | 0.80 | 0.60 | 0.51 | 0.56 | 0.55 | 0.65 | **0.85** | 0.53 | 0.53 | **0.89** | 0.89 |
| eating | 0.73 | 0.55 | 0.50 | 0.50 | 0.50 | 0.51 | **0.78** | 0.50 | 0.50 | **0.79** | 0.82 |
| exercise | 0.69 | 0.59 | 0.50 | 0.53 | 0.50 | 0.53 | **0.74** | 0.50 | 0.50 | **0.75** | 0.77 |
| farming | 0.77 | 0.56 | 0.51 | 0.54 | 0.52 | 0.57 | **0.83** | 0.51 | 0.51 | **0.81** | 0.88 |
| metal | 0.64 | 0.57 | 0.50 | 0.50 | 0.50 | 0.51 | **0.67** | 0.50 | 0.50 | **0.67** | 0.70 |
| still water | 0.70 | 0.54 | 0.51 | 0.53 | 0.51 | 0.52 | **0.76** | 0.50 | 0.50 | **0.71** | 0.81 |
| clouds | 0.78 | 0.77 | 0.70 | 0.74 | 0.74 | 0.75 | **0.80** | 0.65 | 0.74 | **0.79** | 0.84 |
| sunny | 0.60 | 0.67 | 0.65 | 0.67 | 0.62 | 0.60 | **0.73** | 0.59 | 0.57 | **0.72** | 0.78 |

Table 1: Accuracy (% of correctly labeled images) for the SUN dataset. **B** and **U** refers to balanced and unbalanced training data, respectively. All local results to left of double line use $k$ values automatically selected per method and per test instance; all those to the right use a fixed $k$ for all queries. See text for details.

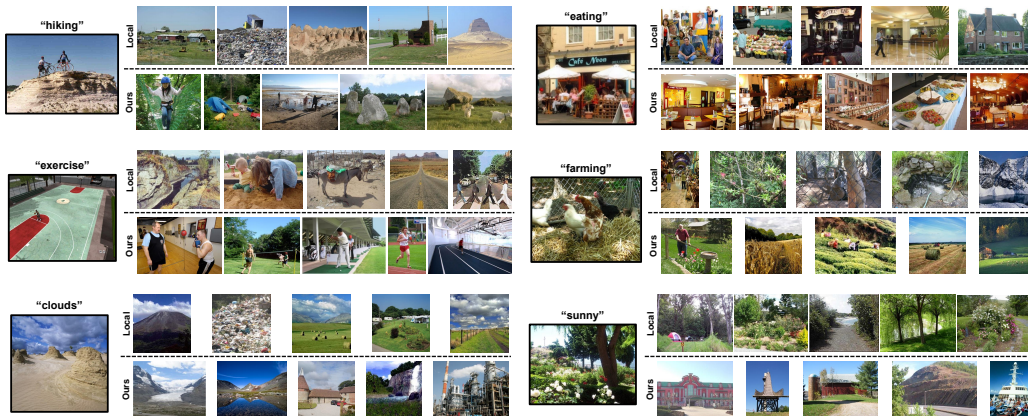

Figure 1: Example neighborhoods using visual similarity alone (Local) and compressed sensing inference (Ours) on SUN. For each attribute, we show a positive test image and its top 5 neighbors. Best viewed on pdf.

ric per cluster, similar to [26, 28], then proceeds as Local+ML. Its results are similar to those of Local+ML (see Supp. file).

The results left of the double bar correspond to auto-selected $k$ values *per query*, which averaged $k = 106$ with a standard deviation of 24 for our method; see Supp. file for per attribute statistics. The rightmost columns of Table 1 show results when we fix $k$ for all the local methods *for all queries*, as is standard practice.[2] Here too, our gain over Local is sizeable, assuring that Local is not at any disadvantage due to our $k$ auto-selection procedure.

The rightmost column, Fix-$k$\*, shows our results had we been able to choose the optimal *fixed $k$* (applied uniformly to all queries). Note this requires peeking at the test labels, and is something of an upper bound. It is useful, however, to isolate the quality of our neighborhood membership confidence estimates from the issue of automatically selecting the neighborhood size. We see there is room for improvement on the latter.

Our method is more expensive at test time than the Local baseline due to the compressed sensing reconstruction step (see Sec. 3.4). In an attempt to equalize that factor, we also ran an experiment where the Local method was allowed to check more candidate $k$ values than our method. Specifically, it could generate as many (proximity-based) candidate neighborhoods at test time as would fit in the run-time required by our approach, where $k$ ranges from 20 up to 6,000 in increments of 10. Preliminary tests, however, showed that this gave no accuracy improvement to the baseline. This indicates our method's higher computational overhead is warranted.

Despite its potential to handle intra-class variations, the Local baseline fails on SUN because the neighbors that look most similar are often negative, leading to near-chance accuracy. Even when we balance its local neighborhood by label, the positives it retrieves can be quite distant (e.g., see "exercise" in Fig. 1). Our approach, on the other hand, combines locality with what it learned about

| Attribute | Global | | Local | | Local+ML | | Ours | Local | Local+ML | Ours | Ours |
|---|---|---|---|---|---|---|---|---|---|---|---|
| | B | U | B | U | B | U | | | $k = 400$ | | Fix-$k$* |
| wing | 0.69 | **0.76** | 0.58 | 0.67 | 0.59 | 0.67 | 0.71 | 0.50 | 0.53 | **0.66** | 0.78 |
| wheel | 0.84 | **0.86** | 0.61 | 0.71 | 0.62 | 0.69 | 0.78 | 0.54 | 0.63 | **0.74** | 0.81 |
| plastic | 0.67 | **0.71** | 0.50 | 0.60 | 0.50 | 0.54 | 0.64 | 0.50 | 0.50 | **0.54** | 0.67 |
| cloth | **0.74** | 0.72 | 0.70 | 0.67 | 0.72 | 0.68 | 0.72 | **0.69** | 0.65 | 0.64 | 0.77 |
| furry | 0.80 | 0.80 | 0.58 | 0.75 | 0.60 | 0.71 | **0.81** | 0.54 | 0.63 | **0.72** | 0.82 |
| shiny | 0.72 | **0.77** | 0.56 | 0.67 | 0.57 | 0.64 | 0.72 | 0.52 | 0.55 | **0.62** | 0.73 |

Table 2: Accuracy (% of correctly labeled images) for the aPascal dataset, formatted as in Table 1

useful neighbor *combinations*, attaining much better results. Altogether, our gains over both Local and Local+ML—20 points on average—support our central claim that learning what makes a good neighbor is not equivalent to learning what makes a good *neighborhood*.

Figure 1 shows example test images and the top 5 images in the neighborhoods produced by both Local and our approach. We stress that while Local's neighbors are ranked based on visual similarity, our method's "neighborhood" uses visual similarity only to guide its sampling during training, then directly predicts which instances are useful. Thus, purer visual similarity in the retrieved examples is *not* necessarily optimal. We see that the most confident neighborhood members predicted by our method are more often positives. Relying solely on visual similarity, Local can retrieve less informative instances (e.g., see "farming") that share global appearance but do not assist in capturing the class distribution. The attributes where the Local baseline is most successful, "sunny" and "cloudy", seem to differ from the rest in that (i) they exhibit more consistent global image properties, and (ii) they have many more positives in the dataset (e.g., 2,416 positives for "sunny" vs. only 281 for "farming"). In fact, this scenario is exactly where one would expect traditional visual ranking for local learning to be adequate. Our method does well not only in such cases, but also where image nearness is not a good proxy for relevance to classifier construction.

**aPascal Results** Table 2 shows the results on the aPascal dataset. Again we see a clear and consistent advantage of our approach compared to the conventional Local baselines, with an average accuracy gain of 10 points across all the Local variants. The addition of metric learning again provides a slight boost over local, but is inferior to our method, again showing the importance of learning good *neighborhoods*. On average, the auto-selected $k$ values for this dataset were 144 with a standard deviation of 20 for our method; see Supp. file for per attribute statistics.

That said, on this dataset Global has a slight advantage over our method, by 2.7 points on average. We attribute Global's success on this dataset to two factors: the images have better spatial alignment (they are cropped to the boundaries of the object, as opposed to displaying a whole scene as in SUN), and each attribute exhibits lower visual diversity (they stem from just 20 object classes, as opposed to 707 scene classes in SUN). See Supp. file. For this data, training with all examples is most effective. While this dataset yields a negative result for local learning on the whole, it is nonetheless a positive result for the proposed form of local learning, since we steadily outperform the standard Local baseline. Furthermore, in principle, our approach could match the accuracy of the Global method if we let $k_K = M$ during training; in that case our method could learn that for certain queries, it is best to use *all* examples. This is a flexibility not offered by traditional local methods. However, due to run-time considerations, at the time of writing we have not yet verified this in practice.

## 5 Conclusions

We proposed a new form of lazy local learning that predicts at test time what training data is relevant for the classification task. Rather than rely solely on feature space proximity, our key insight is to learn to predict a useful neighborhood. Our results on two challenging image datasets show our method's advantages, particularly when categories are multi-modal and/or its similar instances are difficult to match based on global feature distances alone. In future work, we plan to explore ways to exploit active learning during training neighborhood generation to reduce its costs. We will also pursue extensions to allow incremental additions to the labeled data without complete retraining.

**Acknowledgements** We thank Ashish Kapoor for helpful discussions. This research is supported in part by NSF IIS-1065390.

## Footnotes

[1] In our experiments the datasets have binary labels ($C = 2$); in the case of $C > 2$ the local model must be multi-class, e.g., a one-versus-rest SVM.

[2]We chose $k = 400$ based on the range where the Local baseline had best results.

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
