[Supplementary Material]

# Predicting Useful Neighborhoods for Lazy Local Learning
## (Supplementary File)

This supplementary file provides more details on the datasets used in our experiments, additional qualitative examples, and the results from a local metric learning baseline, and more analysis of the learned neighborhoods.

## 1. SUN Attribute Dataset

Figure 1: Sample images from SUN Attribute dataset.

## 2. aPascal Dataset

wing

wheel

plastic

cloth

furry

shiny

Figure 2: Sample images from aPascal dataset.

# 3. Additional qualitative examples

Figure 3: Additional example neighborhoods using visual similarity alone (Local) and compressed sensing inference (Ours) on SUN. For each attribute, we show a positive test image and its top 5 neighbors. Best viewed on pdf.

| SUN | #Pos Sample | Mean $k$ | Local+LML | |
|---|---|---|---|---|
| | | | B | U |
| hiking | 700 | 110 | 0.51 | 0.60 |
| eating | 441 | 121 | 0.50 | 0.50 |
| exercise | 401 | 151 | 0.50 | 0.52 |
| farming | 281 | 120 | 0.51 | 0.53 |
| metal | 1203 | 88 | 0.50 | 0.50 |
| still water | 469 | 67 | 0.50 | 0.53 |
| clouds | 1516 | 108 | 0.70 | 0.72 |
| sunny | 2416 | 84 | 0.64 | 0.63 |

Table 1: Learned neighborhood sizes for SUN (left) and the local metric learning baseline result (right).

| aPascal | #Pos Sample | Mean $k$ | Local+LML | |
|---|---|---|---|---|
| | | | B | U |
| wing | 321 | 122 | 0.59 | 0.66 |
| wheel | 824 | 164 | 0.58 | 0.72 |
| plastic | 616 | 179 | 0.50 | 0.57 |
| cloth | 2584 | 151 | 0.70 | 0.73 |
| furry | 651 | 130 | 0.59 | 0.74 |
| shiny | 1028 | 158 | 0.55 | 0.68 |

Table 2: Learned neighborhood sizes for aPascal (left) and the local metric learning baseline result (right).

## 4. Local metric learning baseline

As discussed in the main text, we also attempted a baseline that uses both local metric learning and local lazy learning. This baseline is just like Local+ML, except that it applies a *local* metric to identify the test instance's nearest neighbors. Following the underlying idea of local metrics developed in [2, 3], we first cluster the training images into $L$ clusters, and then apply metric learning on each cluster. We again use ITML [1] for the metric learning algorithm. Then, for each test image, we determine the closest cluster using Euclidean distance and apply a classifier trained with only its nearest neighbors, as determined using that cluster's respective learned metric. We used $L = 20$ clusters in our experiments, which is the number of object categories in the PASCAL dataset. (Recall that our task is attribute classification, not object classification.)

The rightmost two columns in Tables 1 and 2 show the results for this baseline, which we name **Local+LML** to denote that it uses local learning plus local metric learning. We provide results whether using balanced (B) or unbalanced (U) neighborhoods, in the same format as the main text. We see that Local+LML fares similarly to the Local+ML baseline overall, and underperforms our method for both datasets.

## 5. Analyzing the learned neighborhoods' composition

Tables 1 and 2 show some statistics for the SUN and aPascal datasets, respectively, to help understand how the learned neighborhoods vary. In particular, we list the number of positive instances per attribute that exist in these labeled datasets, as well as the mean auto-selected $k$ values our method chooses for each attribute. Interestingly, while values around $k = 400$ are generally preferred by the Local baseline, our method tends to generate learned neighborhoods that are smaller on average— namely, 106 or 144 on average for SUN and aPascal, respectively.

We find that our predicted neighborhoods tend to contain more unique objects than that of the Local's. Specifically, the ratio of the number of unique classes to the selected $k$ value averages 87 and 35 on SUN and aPascal, respectively, whereas it is only 60 and 28 for Local. They also contain more positives in the unbalanced case—on average 70% positives vs. Local's 13% positives. Therefore, we tend to find images from more distinct object classes, which can help with intra-attribute variations. Furthermore, when comparing our predicted neighbors to Local's, we find that the average Jaccard coefficient is small, 0.05 for SUN and 0.09 for aPascal. This indicates that the learned neighborhoods do deviate from the strict proximity rankings.