[Reviews · NeurIPS 2014]

Submitted by Assigned_Reviewer_6

This paper introduces a system that creates a classifier at test time for a given test image using a set of training examples in a specific neighborhood of the test image. The authors method proposes how to identify the neighborhood with the most informative training examples for the given test image. The authors indicate that their method is well suited to fine-grained image classification tasks.

The authors’ key observation is that while training images that are nearest to the test image may be informative, the most informative set of training images must be found by estimating the informativeness of all members of a neighborhood set together. The main contribution of this paper is a method for estimating the most informative neighborhood set in an online fashion for each new test image.

This method would be particularly useful in classification problems where members of a given class do not often match the prototype of that class. If categories have relatively little intra-class variation, the authors’ method would not improve over traditional offline methods that simply train with the entire available training set. However, in the case of attribute recognition, an attribute category may have a wide visual variety. I found the authors’ method to be an interesting examination of one possible way of handling strong intra-class variation.

The authors make a reference to possible high computation cost during the offline training phase. This is a reasonable sacrifice if the offline training phase only has to happen once. It seems the learned mapping from phase 2 of the pipeline is only for a particular set of categories. If the number of classes in a given domain can be successfully expanded without offline retraining or with rapid retraining, that would make a more convincing argument for this approach.

Lines 202-213 most important part of the paper. The compressed sensing explanation is too condensed, in my opinion.

The neighborhood definition method in lines 233-239 seems like a quick, elegant solution. Using K different neighborhoods seems not too computationally demanding. I’d have liked to see a demonstration of on average how different the K candidate neighborhoods are for a given test example.

Line 251 is an insightful observation. The authors’ method is particularly well suited to highly imbalanced datasets. As highly imbalanced training sets are common in fine-grained vision research, and an increasingly common problem in real-world vision, the problem the authors address in this paper is of interest to the fine-grained visual categorization community.

It would have been good to see results over a larger number of visual categories. Especially categories that have strong intra-class variations and rare positive examples (the ‘scary’ class from the SUN attribute dataset for example).

The authors’ method shows significant classification gains over the traditional vision pipeline, especially for those categories that have strong intra-class variation.
Summary: This topic is of great interest to the fine-grained vision community. This paper has a clever solution, and discussion about the authors' method will be valuable to the community.

Submitted by Assigned_Reviewer_16

This paper proposes a method to predict, given a novel test instance, a useful subset of examples to train a local classifier. Existing methods in the space of lazy local learning typically fix the size of the selected subset and select relevant training instances based on the proximity (nearest neighbors) to the test example. By contrast, the proposed method estimates the required size of the relevant subset while allowing for deviation from a strict nearest neighbor ranking.

The method consists of three main steps. The first (offline) is to generate a training set of instance-neighborhood pairs (obtained by distance-weighted sampling of neighbordhoods of varying sizes and for each training intance, heurestically selecting the best performing such pairs). The second (also offline) is to rely on the multi-label classification framework of [19] to learn a mapping between an instance and its useful neighborhood (reperesented as an indicator vector of confidences over the entire training set). The third (online) applies the learned mapping to a novel test intance to predict an indicator vector of confidences: from this vector, candidate neighbordhoods, of varying sizes, are generated via confidence-weighted sampling and used to train a number of local classifiers. Finally the prediction of the most confident local classifier is retained.

This paper is extremely well written. The ideas are presented very clearly. The authors constantly situate the paper and the proposed methods well within the existing body of works. The result is a paper that reads well and highligths the important differences with respect to existing works.

My main concern at this juncture is the fairness of the experimental evaluation. In general, the reviewer finds it surprising that the local baselines perform so poorly (almost consistently at chance) on the SUN dataset given that it is indeed perfectly well suited for lazy learning. However, beyond this, in the current setup it is unclear wether the gains with respect to the local baselines are due to the learning of relevant neighborhoods (steps 1 and step 2) or whether they are simply due to choosing the most confident local classifier out of a presumably large (how large is it?) set of trained classifiers (the final step in step 3).

In other words, the reviewer believes a key and important baseline is missing here. This baseline would consist of applying step 3 on a confidence indicator vector (y^_q) whose entries are proportional to the distance (eucledian or learned) to the test query. In this manner, we can evaluate whether the reported gains are due to the learned y^_q or whether they are due to the trial and error tests of step 3, in which case the contribution of the paper is very small.

In addition, the reviewer does not believe that using the auto-selected k from the proposed method to evaluate the local baseline is fair. The current reported local baselines should have their own auto-selected k values. If this is the case, please clarify.

Also, again with regards to experiments, it would be very important (even critical) to make explicit the exact number of positive training instances per data set and per attribute as well as the distribution of k per attribute so that we may get an understanding of the proportion of positives that are selected.

Finally, the reviewer would like a motivation for the computation of k* in line 175 via an averaging of probabilities (in general probabilities cannot be averaged) for step 1. Also, how is the Platt calibration performed exactly? Finally, are the local classifiers in step 3 calibrated? If so, how? How many such local classifiers are used?

Summary: A key baseline is missing that would allow the reviewer to understand whether the reported gains are truly arising due to the learning of useful neighborhoods. A number of very important questions (see above) need to be answered as well.

UPDATE: The rebuttal has addressed my main concern and I have changed my assessment accordingly. The reviewer is convinced that useful neighborhoods are being learned. A few recommendations:

1) I would clarify what the baselines are exactly in both section 3.3 and section 4 'Baselines and setup'. Confusion will be too easy otherwise in the reviewer's opinion.

2)Include as promised the number of instances per attribute and the distribution of K per attribute. This is imporant as it will give the reader an idea on the proportion of selected examples.

3)Describe specifically how the Platt calibration is done: cross-validation, ect..

Submitted by Assigned_Reviewer_30

The paper identifies the problem that the local neighborhood typically used by lazy local learning approaches for (visual) classification is frequently not sufficient to learn a good model. To attack the problem the paper proposes an approach which generates good neighborhoods and learns a mapping from instances to good neighborhoods in an offline phase. At test time the neighborhood for the test instance is predicted and classifiers are learned on this neighborhood to predict the category of the test instance.

Positive aspects:
- The paper is clearly written and easy to follow.
- The paper attacks the important and difficult problem of recognizing very multi-modal, diverse categories. Although the approach is applied only to attribute classification in images, it is not specific to images and could to my understanding also be useful in many other domains.
- The paper provides an extensive experimental evaluation on two challenging attribute datasets and also provides qualitative results.
- The authors clearly discuss related work and provide the differences to their work.

Weaknesses:
- It seems that the approach works only very well for categories that are indeed *very* multi-modal and diverse as for attribute recognition where an attribute spans over several objects. Already in the case of more similar objects such is the aPascal dataset, standard global classification approaches win. I expect the method to clearly loose and standard object recognition challenges.
- It is not clear why only a subset of attributes is selected for evaluation rather than evaluating on all.
- Supplemental work could have shown more qualitative results rather than just the datasets. It also shows only 7 of the 8 selected attribute categories for the SUN dataset.

Points which I would like to see discussed in the rebuttal
- The approach seems very powerful in the case when the estimated categories are very diverse and multi-modal. Maybe the authors could discuss how to integrate global information into their model (at the same time while using the proposed neighborhoods), which would allow to also recognize more homogenous categories better. Maybe this would yield improved performance, especially on aPascal? Could one e.g. learn a global model and use domain adaptation techniques / fine-tuning for deep learning to adapt the model to the specific neighborhood?
- Both aPascal and SUN, also have to my knowledge object category labels in addition to the attribute labels. To better understand what the approach is doing: How can the neighborhoods be characterized, do they typically consist of a small subset of the object categories (this is what Figure 1 suggests)?
Summary: The paper presents a novel and interesting approach for image classification in scenarios where the category to be separated are very multi-modal and unbalanced. The experimental evaluation indicates benefits over other lazy local learning methods but it remains questionable if the method will be of great impact due to the computational complexity and limited performance compared to standard global classification approaches.

Submitted by Meta_Reviewer_8

AC's review

The general idea of trying to predict useful neighbours for learning
is interesting but I believe that the method as it stands is
fundamentally flawed. By its nature compressed sensing can only ever
return points that have appeared in at least one of its training
neighbourhoods. This might be a useful means of eliminating points
deep in the interiors of their classes that are never useful for
discrimination, but it has two obvious flaws:

- Firstly, the nearest neighbours of the given test point might be
excluded by chance, especially if it is in a sparsely populated region
and the pre-training time is limited. This seems counterproductive --
the (balanced) near neighbours of the test point should probably
always be included in the final local classifier, whatever else
is.

- Secondly, to counter this the pre-training will need to sample a
large fraction of the training dataset, so the compressed sensing
basis will essentially include the entire training set. This will make
it bulky and very slow to evaluate at test time. Probably so much so
that other "simpler" strategies would be preferable, such as the
pre-training-free one of repeatedly sampling balanced sets of
neighbours of the test point at test time, training classifiers over
them, and either selecting the best classifier (based e.g. on its
performance on held-out neighbours or its certainty on the test point)
or applying an ensemble method such as voting, boosting or ensemble
SVM over the sampled set of classifiers. Compressed sensing is likely
be so slow that a good deal of sampling can be done at this stage in
place of it.

Moreover, the paper failed to convince me that "good neighborliness"
intrinsically requires a complex set-level relevance metric such as a
full compressed sensing expansion in the first place, and not just a
simple scalar one such as individual closeness to the decision
boundary of a global classifier (or an ensemble of local ones, etc).
This is a critical point because there are many successful active
learning methods that take the scalar viewpoint, each with its
corresponding relevance metric. I would expect running the above local
sampling strategy under a suitable individual relevance metric to give
good results, so I think we need significantly stronger evidence of
the need for the set-level viewpoint before accepting compressed
sensing as a useful strategy for this kind of application.

In summary, to make the conclusions believable, we need to see better
baselines including one or more of the above "sample sets of
relevant-looking (scalar metric) local neighbours at test time and
take the best (best combination) classifier" strategies, with detailed
performance figures and analyses including comparisons of training and
test run times and/or performance figures at equal run-time
effort. We also need to see at least one baseline based on *local*
metric learning. The active learning literature is highly relevant to
this program and it needs to be cited and discussed much more
thoroughly.
Summary: This is an interesting attempt to use compressed sensing to define
suitable neighbourhoods for training ad hoc local classifiers at test
time. However I felt that the method was flawed and the experimental
results failed to convince me owing to lack of testing against some of
the most obvious competitors.
Author Feedback
Author rebuttal: Thanks for the valuable feedback.

AR16’s comments indicate important misunderstandings about experiments.

==AR16

= “Main concern…Key baseline is missing...Unclear whether gains due to learning relevant neighbors”
We do show this baseline. The baseline AR16 requests is exactly what we implemented for the Local and Local+ML baselines. They sort the labeled instances by Euclidean (or learned) distance to the test query, then apply what AR16 refers to as “step 3”. So, our gains *are* due to the learned y_q, not step 3. We realize now that Sec 4 does not make it explicit enough that “step 3” (i.e., selecting k automatically based on the most confident classifier) is also applied for the Local baselines---independently of our method’s results. That is, the Local baselines choose their own k values. We will clarify.

= “Using the auto-selected k from the proposed method to evaluate the local baseline is not fair.”
Right, that would not be fair. The Local baseline does *not* use the auto-selected k from the proposed method. It does its own auto-selection. So, we did implement it just as AR16 suggests. See L365-73. We will clarify in Line 312.

= Surprising the local baselines do so poorly on SUN
We discuss why in L374-405. We actually made substantial efforts to optimize the Local baseline’s results: 1) we report both the balanced (B) and unbalanced (U) scenarios, 2) we add metric learning (ML), 3) we try both auto-selected k values and a fixed k value that was best for the baseline (see RHS of tables). However, even under all these variations, it remains inferior to our approach.

= Make explicit the num pos instances and distribution of k
In order of attributes in Tables 1 & 2, the #positive instances are 700,441,401,281,1203,469,1516,2416 (out of 14,340 img for SUN), and 321,824,616,2584,651,1028 (out of 6,440 img for aPascal). See L366 and 410 for overall k distributions; we’ll add per attribute/method.

= k* computation
The idea is to smooth the estimated probability using the samples for that value of k, each of which favors near points but varies in its composition. It can be seen as a simple but effective way to take a vote per candidate k, avoiding relying solely on any one sample.

= How is Platt done?
We use LIBSVM, fitting a sigmoid to the classifier outputs of the neighborhood points.

= Are local classifiers in step 3 calibrated? How many are used?
No. Step 3 yields a single local classifier (L175-8); they are not combined.

==AR30

= How to integrate “global” info: domain adaptation?
An interesting idea – one could regularize the local model to be similar to the global model. As AR30 notes, this would likely boost results when the data is more homogenous. A practical difficulty, however, is to dynamically adjust the regularization term’s impact as a function of the query and its neighbors; “over-regularizing” would eliminate the gains our method offers wrt Global for more difficult heterogeneous data. Also, note that with k_K = M, our method can *learn* to revert to the global approach for suitable queries (as discussed in L418-22).

= Characterize learned neighborhoods
They tend to contain more unique objects than Local’s. Specifically, the ratio of the number of unique classes to the selected k value averages 87 and 35 on SUN and Pascal, resp, whereas it is only 60 and 28 for Local. They also contain more positives in the unbalanced case (L397) – on average 70% positives vs Local’s 13% positives. So we tend to find images from more distinct object classes, which can help with intra-attribute variations.

= Performance compared to standard global classification
Our approach consistently outperforms the Global baseline for all 8 classes in SUN. This is a strong result. In contrast, the standard Local method is barely better than chance. Furthermore, on aPascal, the pseudo “upper bound” for our method (see Ours Fix-k*) outperforms Global in half of the classes tested, indicating the potential on that dataset with better neighborhood size selection (see L369-73).

= Works well for categories that are multi-modal
We do expect local learning methods, including ours, to be most beneficial for the challenging scenario of high intra-class variation. Our empirical study of attributes aims to capitalize on this. Whether or not the algorithm can also boost other object recognition tasks---or non-vision classification tasks---remains to be seen.

= Why subset of attributes for evaluation?
We took a random subset of attributes per dataset. This was purely due to the computational cost to run experiments in all the variations shown for our method and the baselines. Sec 3.4 summarizes our run-time.

= Supp could show more image ex
Sure, we can add more like Fig 1 in the main paper.

= Supp img for 7 attributes
We inadvertently omitted “sunny”. We’ll fix.

==AR6

= Compressed sensing explanation too condensed
Got it, we will expand.

= How different are the K candidate neighborhoods?
Comparing our predicted neighbors to Local’s, we find the average Jaccard coeff is small, 0.05 for SUN and 0.09 for aPascal. Note whereas during training the candidate neighborhoods vary in size and composition, at test time (L233-9) they vary only in size, since the neighbor ranking is “read off” directly from y_q.

= Would be good to see attributes with rare positives and high intra-class variation, like “scary”
Good idea, thank you. We simply randomly chose the classes to consider. But it makes sense to focus future experiments on these most challenging cases, where our method should in principle provide even larger gains.

= Rapid retraining for new class?
As typical of most any classification scheme, we presume the categories of interest are known during training. It will be interesting future work to consider incremental training extensions for multi-label classification via compressive sensing.